# Geospatial distribution and predictive modeling of onchocerciasis in Ogun State, Nigeria

Olabanji Ahmed Surakat[1]☯*, Ayodele S. Babalola[2☯], Monsuru A. Adeleke[1], Adedapo O. Adeogun[2], Olufunmilayo A. Idowu[3], Sammy O. Sam-Wobo[3]

1 Faculty of Basic and Applied Sciences, Department of Zoology, Osun State University, Osogbo, Nigeria, 2 Department of Public Health and Epidemiology, Nigerian Institute of Medical Research, Yaba, Lagos, Nigeria, 3 Department of Pure and Applied Zoology, Federal University of Agriculture, Abeokuta, Nigeria

☯ These authors contributed equally to this work.
* olabanji.surakat@uniosun.edu.ng

**Data Availability Statement:** All relevant data are within the paper and its Supporting information files.

## Abstract

Onchocerciasis caused by infection with *Onchocerca volvulus* is a disease of public health importance and is highly associated with disability. As Nigeria is aiming at eliminating onchocerciasis by 2030, there is a need to develop newer tools to map disease prevalence and identify environmental factors driving disease prevalence, even in places that have not been previously targeted for preventive chemotherapy. This study produced predictive risk-maps of onchocerciasis in Ogun State. Georeferenced onchocerciasis infection data obtained from a cross-sectional survey at 32 locations between March and July 2015 together with remotely-sensed environmental data were analyzed using Ecological Niche Models (ENM). A total of 107 field occurrence points for *O. volvulus* infection were recorded. A total of 43 positive occurrence points were used for modelling. ENMs were used to estimate the current geographic distribution of *O. volvulus* in Ogun State. Maximum Entropy distribution modeling (MaxEnt) was used for predicting the potential suitable habitats, using a portion of the occurrence records. A total of 19 environmental variables were used to model the potential geographical distribution area under current climatic conditions. Empirical prevalence of 9.3% was recorded in this study. The geospatial distribution of infection revealed that all communities in Odeda Local Government Area (a peri-urban LGA) showed remarkably high prevalence compared with other LGAs. The predicted high-risk areas (probability > 0.8) of *O. volvulus* infection were all parts of Odeda, Abeokuta South, and Abeokuta North, southern part of Imeko-Afon, a large part of Yewa North, some parts of Ewekoro and Obafemi-Owode LGAs. The estimated prevalence for these regions were >60% (between 61% and 100%). As predicted, *O. volvulus* occurrence showed a positive association with variables reflecting precipitation in Ogun State. Our predictive risk-maps has provided useful information for the elimination of onchocerciais, by identifying priority areas for delivery of intervention in Ogun State, Nigeria.

**Funding:** The author(s) received no specific funding for this work.

**Competing interests:** The authors have declared that no competing interests exist.

## Introduction

Onchocerciasis also known as river blindness is a tropical disease caused by the filarial nematode *O. volvulus* and transmitted through repeated bites of *Simulium* blackflies. The epidemiology of the disease is greatly influenced by the presence of adequate vector breeding sites [1]. The vector of onchocerciasis breeds in fast-flowing rivers where they attach to rocks, plants and suspended particles, or smaller river habitats of freshwater crab (*Potamonautes* spp.) [2,3]. Members of the *Simulium damnosum* complex have however been implicated as the only vectors of human onchocerciasis in West Africa [4]. Clinical manifestation of the diseases could either be ocular or dermal with symptoms such as nodules, itching, skin thickening, visual impairment, and blindness [5].

Onchocerciasis is of public health concern in 31 sub-Saharan African countries. However, Yemen, South America, Ecuador, Colombia, and Mexico have been verified by World Health Organization (WHO) to have successfully interrupted the transmission of *O. volvulus* [1]. Nigeria has one of the heaviest burdens of onchocerciasis in the world, accounting for almost one-third of the global prevalence [6,7]. Nodules removal (nodulectomy), vector control and mass drug administration (MDA) with ivermectin are some of the interventions that have been implemented in the control of morbidity of onchocerciasis and interruption of transmission of parasite with varying degrees of success [1,8]. In most areas of Africa, annual community-directed treatment with ivermectin (CDTI) is the primary intervention for control of onchocerciasis, with the exception of a few foci where semi-annual treatment is implemented [1]. In Ogun state, Nigeria, eight local government areas (LGAs) have been reported to be endemic for onchocerciasis, with two suitable vector habitats (Ogun and Osun rivers) with submerged rocks and vegetation, traversing several of the communities in these LGAs [7]. Following rapid epidemiological mapping of onchocerciasis (REMO), control strategy using CDTI have been implemented in Ogun state since 2004 [9], however, despite the intensified global efforts to eliminate onchocerciasis and the success achieved with MDA in Ogun state, the presence of microfilaria (mf) observed amongst adult population is a reflection that mass treatment may not really be effective [7], or a result of varying degrees of acceptability of medicines in the communities [10]. In line with the elimination targets set by WHO, there is a need to produce a model-based estimates of level of risk of Onchocerciasis across the communities or regions of Ogun state and this will invariably help policy makers to know where intervention strategies may require modifications and intensification. In this study, the geographical information system (GIS) and remote sensing (RS) technology were employed utilizing *O. volvulus* infection data from a cross-sectional survey across 8-endemic LGAs in Ogun State Nigeria, to produce model-based estimates of infection risk for the whole of the State.

Advancement in geospatial technology have made it possible to conduct disease mapping, surveillance, outbreaks prediction and evaluating infectious diseases spread pattern, particularly epidemic or pandemic prone-diseases [11–13]. Geospatial technology is now being adopted by experts in the public health field and decision makers to make visualization and analytical tools in order to carry out disease control programs in affected and/or suspected regions. It can also be used to perform analysis and predictions that were previously technologically impossible [13]. This technology is valuable as it is helpful in generating maps which are useful in comprehending the geographical distribution of disease for case frequency study, disease mapping and disease correlation with environmental factors and so on. These geospatial tools can forecast infection across a vast geographical zone by integrating a range of possible disease drivers which include remotely sensed climatic and environmental data in addition to the applicable socio-demographic information to advance model predictions [14].

In Nigeria, onchocerciasis programme has progressed from control to elimination in line with the objective of the WHO-Neglected Tropical Diseases (NTD) 2030 roadmap. To achieve elimination, there is a need to develop newer tools to map disease prevalence, identify socio-demographic and environmental factors driving disease prevalence and to forecast estimates of infection risk even in places that have not been previously targeted for preventive chemo-therapy. In this study, we mapped the distribution of onchocerciasis (based on rapid diagnostic test (RDT) and skin snip data for mf prevalence) at community and local government level in Ogun state while also broadening the geostatistical methods to consider and integrate important environmental inferential covariates. The objective of this study was to present a geospatial map of onchocerciasis infection in Ogun state, identify the major environmental factors that correlate with onchocerciasis infection and to produce model-based estimates of infection risk of onchocerciasis for the whole of Ogun state.

## Materials and methods

### Study area, design and population

This study was conducted in Ogun State, Nigeria (Fig 1). Details of the study area, design and population surveyed have been described elsewhere [7]. In brief, the study was carried out

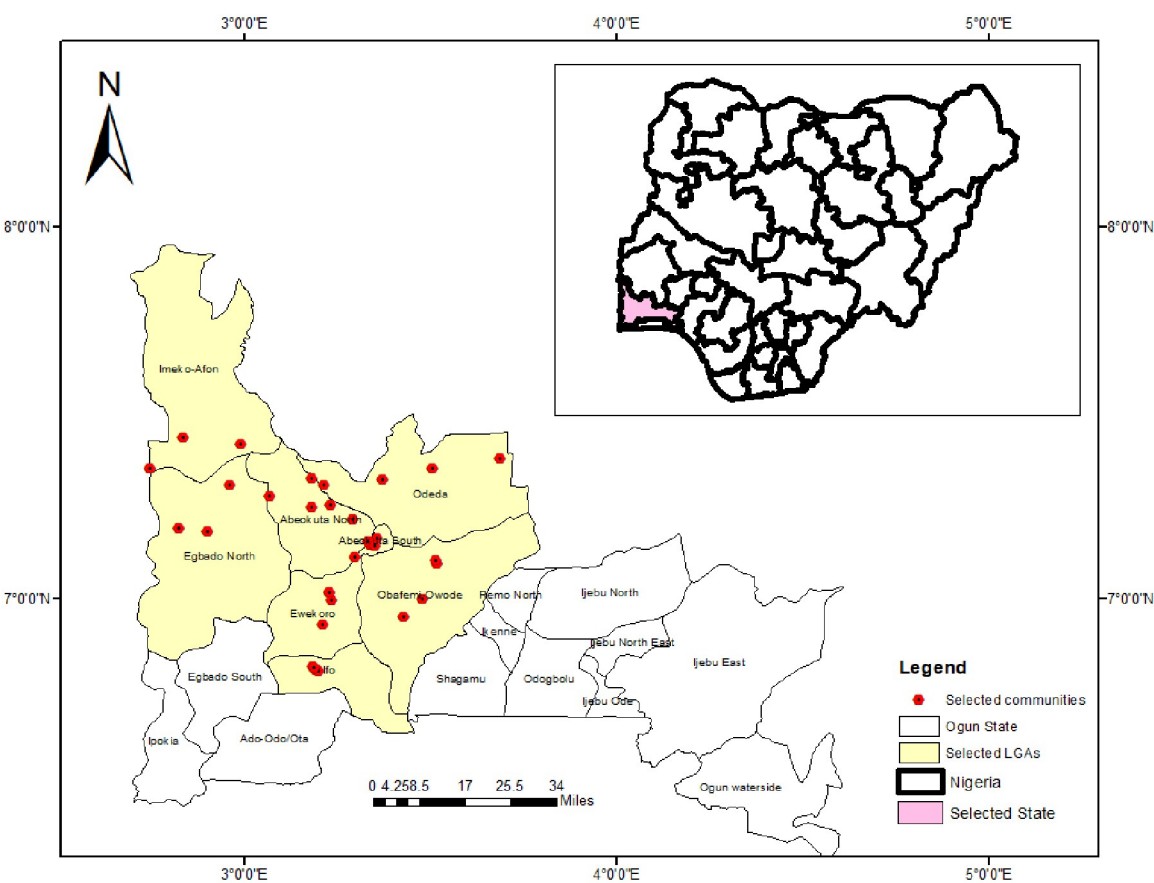

**Fig 1. Map showing the study locations and LGAs in Ogun State.** This figure was created by the authors in R programming software (R version 4.1.2, Vienna, Austria). Available at https://www.R-project.org/. The Nigerian shapefile was obtained from World BankDataCatalog (https://data.humdata.org/dataset/geoboundaries-admin-boundaries-for-nigeria) an Open license standardized resource of boundaries (i.e., state, county) for every country in the world.

between March and July 2015 during the wet season. We designed a cross-sectional survey, and employed a systematic grid sampling method in the selection of communities to ensure an unbiased representation across the selected endemic LGAs [7]. A total of 3,895 children and adults, from 32 spatially selected communities participated in the study [7]. In each community, all households and their occupants were considered eligible for participation and invited to participate in the study. Only participants who gave their consent and were willing to participate were included in this study.

## Onchocerciasis infection data

The field and laboratory procedures have been previously described in [7]. In brief, both RDT and skin snip methods were employed to determine prevalence of onchocerciasis. For the RDT, participants were serially arranged in a census form and the corresponding number was written on each Alere™ SD Bioline IgG4 test kit (Abbott Laboratories, Abbott Park, IL, USA) cassette using a permanent marker for easy identification. The kit was used according to the manufacturer's instruction [6]. For the skin snip examination, two bloodless skin biopsies of each participant were obtained from the left and right posterior iliac crest with the aid of a 2 mm corneoscleral punch (Holth and Modified Walser) and placed on a glass slide with a drop of saline water. The tissues in the slides were examined under a x40 binocular microscope after 30 min for micro-filarial manifestation. Each skin tissue was incubated in a 96-well microtitration plate containing a normal saline solution and when the column was full the wells were covered with a transparent adhesive film and kept for re-examination within 24 h. Mf results were expressed as positive (mf present) or negative (mf absent); mf prevalence was expressed as number of persons positive divided by the total number of persons examined [7,15].

## Environmental data

The climatic variables such as temperature and precipitation influenced global and meso scales and topographic variables such as altitude and aspect that likely affects species distributions at meso and topo-scales while land-cover variables like percent canopy cover influence distributions at the micro-scale [16,17]. Hence, the use of the climatic and topographic variables in the prediction of distributions of *O. volvulus* infection was used in Ogun State.

To determine the possible distribution of *O. volvulus* infection in Ogun State using the prediction model, a total of 107 positive occurrences for *O. volvulus* infection were recorded from the field study. The autocorrelation problems were addressed by eliminating redundant presences on the scale of the bioclimatic variables used in each $1 \times 1$ km grid [18]. In addition, records for spatial autocorrelation were screened in ArcGIS 10.7.1 using average nearest neighbor analyses to remove spatially correlated data points [19,20]. After this selection, a total of 43 *O. volvulus* positive occurrence points were used to create a prediction model. We considered 19 environmental variables as potential predictors of the target species habitat distribution [21,22]. These variables were chosen based on their biological relevance to the target species distributions and other habitat modeling studies [17,23–26]. Nineteen bioclimatic variables, biologically more meaningful to define eco-physiological tolerances of a species [27,28], with a 30 arc-second spatial resolution (about. 1 km$^2$) were downloaded from the WorldClim database (http://www.worldclim.org/) [29].

Then we utilized the standard geographic coordinates in decimal degrees (to four decimal places) in WGS 84. Then we spatialized and checked the geographic coordinates on Google Earth. After downloading the climatic files (covering the period 1950–2000), the Nigeria layer was extracted by using a boundary mask. After that, extracted files were converted to ASCII format via using ArcGIS 10.7.1 software to be use later with Maxent software.

**Table 1. Environmental variables used for modeling the potential distribution of Aedes spp. in the present study.**

| No | Variable | Code/Unit | Source |
|----|----------|-----------|--------|
| 1 | Annual mean temperature | Bio1˚C) | WorldClim |
| 2 | Mean Diurnal Range (Mean of monthly (max temp—min temp)) | Bio2˚C) | WorldClim |
| 3 | Isothermality | Bio3˚C) | WorldClim |
| 4 | Min Temperature of Coldest Month | Bio6˚C) | WorldClim |
| 5 | Mean Temperature of Wettest Quarter | Bio8˚C) | WorldClim |
| 6 | Annual Precipitation | Bio12(mm) | WorldClim |
| 7 | Precipitation Seasonality (Coefficient of Variation) | Bio15(mm) | WorldClim |
| 8 | Precipitation of Coldest Quarter | Bio19(mm) | WorldClim |

All combinations of the 19 environmental variables have been tested for multi-collinearity through the calculation of R-squared in linear regression analysis in R software ver. 4.1.2. In this study, because some of these bioclimatic variables were strongly correlated ($R^2 \geq 0.7$), only those variables that showed little correlation with other predictors were retained; following [30,31]. A total of 8 environmental variables were selected in this study ($R^2 < 0.7$); Annual Mean Temperature (bio 1), mean diurnal range (max. temp–min. temp) (bio2), Isothermality (bio 3), min temperature of coldest month (bio 6), mean temperature of wettest quarter (bio 8), Annual Precipitation (bio12), Precipitation seasonality (Coefficient of variation) (bio 15), and precipitation of coldest quarter(bio19) (Table 1).

## Modeling procedure

The modeling technique maximum entropy distribution or Maxent were used in this study; which has been found to be the most effective among several different modeling methods [22,32,33], and may continue to be effective even with small sample sizes [21,25,34–36]. For the study area, it only requires species presence data (not absence) and environmental variable (continuous or categorical) layers. We used the freely available Maxent software, version 3.3.3, which generates an estimate of the probability of the presence of the species that varies from 0 "unsuitable" to 0.99 "best habitat suitability". ASCII files of the 8 selected environmental variables and a CSV file of species presence coordinates in decimal degrees were used to create the module. Maxent's performance was assessed using a threshold independent Receiver-Operating Characteristic (ROC) analysis and Area Under Receiver- Operating Characteristic Curve (AUC) values (0.5 = random to 1 = perfect discrimination). The algorithm either runs 1000 iterations of these processes or continues until convergence is reached (threshold 0.00001).

For the model, the relative importance of each environmental predictor was evaluated using the percentage contribution of the Jackknife test, which is the best index for small sample sizes [25]. The default logistic output format was chosen, i.e. related to the probability of suitable conditions, ranging from 0 to 1. A total of 75% of the location point data were used for training, and the remaining 25% to test the predictive ability of the model, in addition 10 replicates were considered. Average and Standard deviation values for training and test AUC for the 10 models were extracted from the Maxent text result output. The ASCII output map for the average model for the target species was loaded in ArcGIS 10.7.1 where the prediction models of habitat suitability were divided based on Choudhury et al. (2016) into 5 classes; very low (0–0.1), low (>0.1–0.2), moderate (>0.2–0.4), high (>0.4–0.6) and very high (>0.6) using natural breaks in the symbology tools to produce the habitat suitability model picture [37].

### Ethical clearance and permission

Ethical approval for the study was obtained from the Ethics Review Committee (ERC) of Ogun State Hospital Management Board, Abeokuta (trial registration number: SHA/RES/VOL.2/153). Informed consent forms were duly signed by voluntary participants and by parents or guardians of child participants during the skin snip and Ov16 RDT study. Consent to participate in either skin snips, finger prick or both was contained in a questionnaire administered to each participant prior to the study. Participants were allowed to freely decide their preference without coercion. All methods including recruitment of participants, collection of participant's data and samples, laboratory analysis and data management were performed in accordance with the 1964 Declarations of Helsinki.

## Results

### Data summaries

A total of 3,895 infection data was included in the survey. The demographic characteristics of the study population have been described elsewhere [7]. However, Table 2 summarized the prevalence of *O. volvulus* infection among the examined participants. In short, an overall prevalence of 9.3% (95% confidence interval (CI) 7.2, 11.4) was recorded for *O. volvulus* infection. The geographical distribution of the empirical prevalence for *O. volvulus* infection is presented in Fig 2. The geospatial distribution highlighted higher prevalence in Odeda LGA, Abeokuta North and Obafemi Owode LGA compared with the other LGAs. Equally, *O. volvulus* infection was prevalent at all the selected communities in Abeokuta North and Odeda LGAs while only two communities were prevalent for *O. volvulus* infection at Obafemi-Owode and Imeko-Afon LGAs respectively. A zero percent prevalence was recorded at Yewa North LGA. A high proportion of the participants, 3374(87.3%) have never taken a single dose of ivermectin at the time this study was conducted.

### Geospatial distribution of onchocerciasis by age and sex across the selected LGAs

The distribution of *O. volvulus* infection by age and sex is presented in Fig 3. The results showed that there was no regular pattern of infection with respect to age across all the LGAs. For instance, prevalence was generally higher ($p<0.05$) among adults (25 years and above) compared with those below 25 years at the two highly onchocerciasis endemic LGAs (Odeda and Abeokuta North), while there seems to be no difference ($p>0.05$) in prevalence of *O. volvulus* infection with respect to age across the other LGAs (Fig 3).

**Table 2. Summary of Onchocerciasis prevalence across the selected LGAs.**

| LGAs | No. Examined | No. infected | Prevalence (%) | 95% CI |
|---|---|---|---|---|
| Odeda | 442 | 138 | 31.2 | 27.2–35.2 |
| Abeokuta North | 440 | 82 | 18.6 | 15.7–21.5 |
| Yewa North | 459 | 1 | 0.2 | -0.1–0.5 |
| Obafemi Owode | 333 | 35 | 10.5 | 9.3–11.7 |
| Imeko | 634 | 66 | 6.6 | 4.9–8.3 |
| Ewekoro | 501 | 19 | 3.4 | 2.6–4.1 |
| Abeokuta South | 587 | 21 | 3.6 | 3.1–4.2 |
| Ifo | 499 | 2 | 0.4 | 0.3–0.5 |
| | 3985 | 364 | 9.3 | 7.2–11.4 |

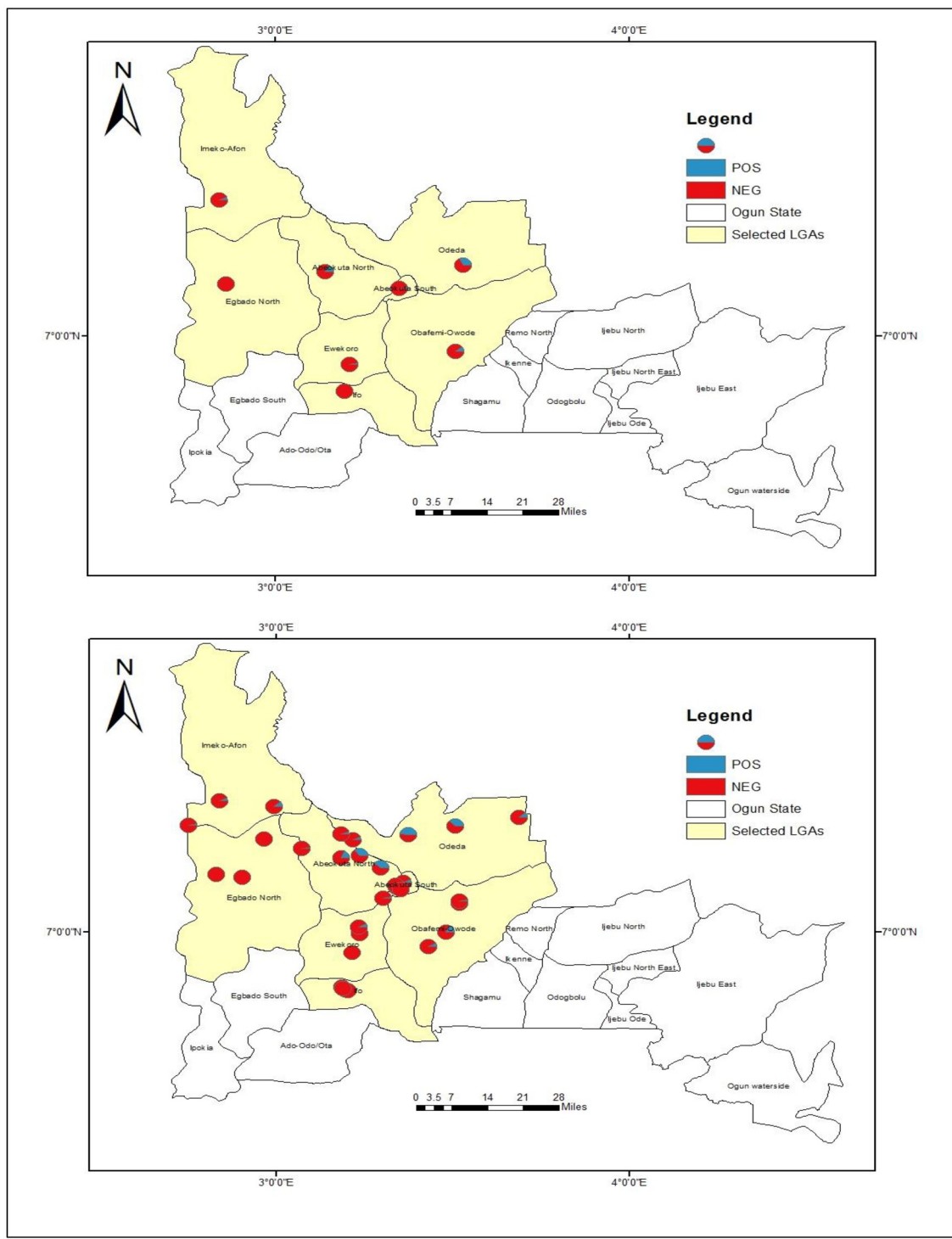

**Fig 2. Geospatial distribution of onchocerciasis at LGA and community level.** This figure was created by the authors in R programming software (R version 4.1.2, Vienna, Austria). Available at https://www.R-project.org/. The Nigerian shapefile was obtained from World BankDataCatalog (https://data.humdata.org/dataset/geoboundaries-admin-boundaries-for-nigeria) an Open license standardized resource of boundaries (i.e., state, county) for every country in the world.

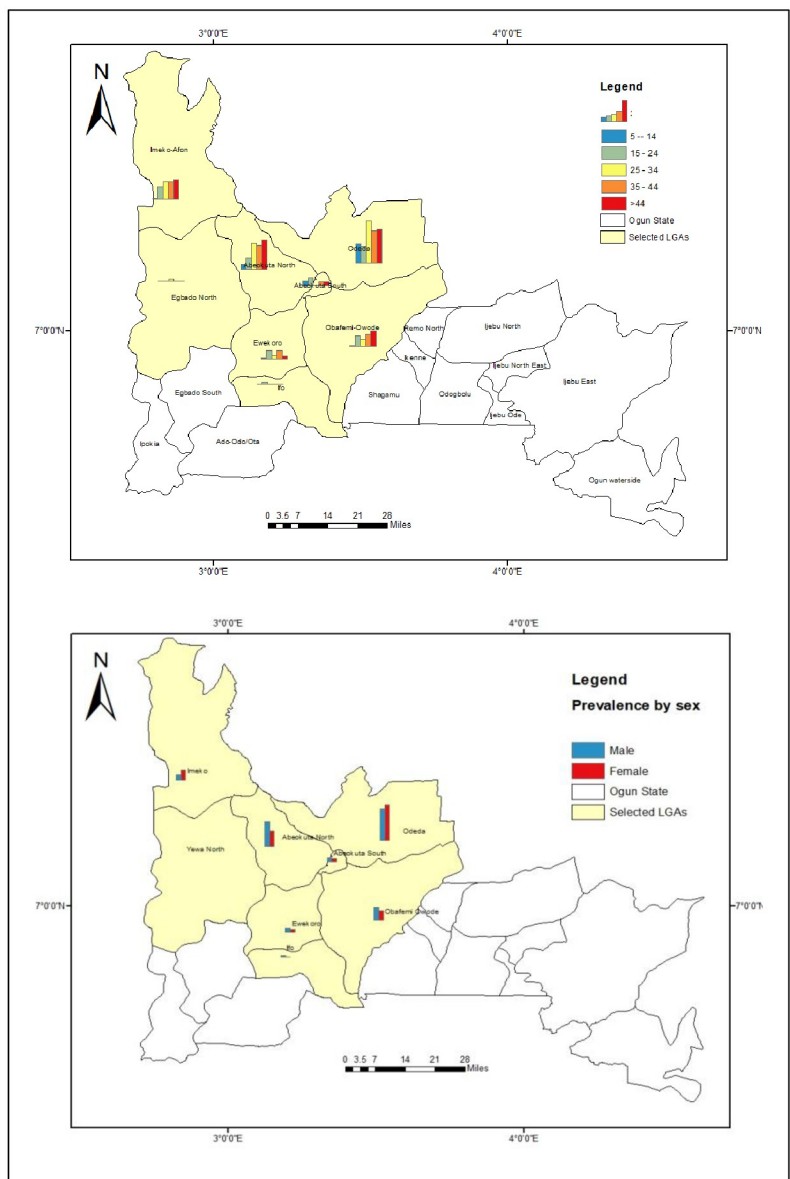

**Fig 3. Geospatial distribution of onchocerciasis by age and sex in the selected LGAs.** This figure was created by the authors in R programming software (R version 4.1.2, Vienna, Austria). Available at https://www.R-project.org/. The Nigerian shapefile was obtained from World BankDataCatalog (https://data.humdata.org/dataset/geoboundaries-admin-boundaries-for-nigeria) an Open license standardized resource of boundaries (i.e., state, county) for every country in the world.

In the same vein, there was no observable pattern in prevalence of *O. volvulus* infection with respect to gender across all the LGAs. Prevalence was higher among females in some LGAs (Imeko-Afon and Odeda) while it was higher in males in other LGAs (Abeokuta North, Ewekoro, Abeokuta South, Ifo and Obafemi-Owode) (Fig 3).

### Logistic regression model for *O. volvulus* infection risk

Except for sex, all other covariates, in Table 3 (LGAs and Age) were significant predictors of *O. volvulus* infection in the models. Prevalence risk of *O. volvulus* infection was highest among

**Table 3. Logistic regression model for predictors of *O. volvulus* infection among the study participants.**

| Variables | COR (95% CI) | p-values | AOR (95% CI) | p-values |
|---|---|---|---|---|
| *LGAs* | | | | |
| Odeda | 112.8 (27.7–458.9) | <0.0001* | 86.0 (21.1–351.3) | <0.0001* |
| Abeokuta North | 56.8 (13.9–232.3) | <0.0001* | 52.0 (12.7–213.1) | <0.0001* |
| Yewa North | 0.5 (0.1–6.0) | 0.618 | 0.6 (0.1–6.3) | 0.642 |
| Obafemi Owode | 29.2 (7.0–122.2) | <0.0001* | 35.2 (8.4–147.9) | <0.0001* |
| Imeko—Afon | 17.6 (4.2–73.2) | <0.0001* | 19.6 (4.7–81.3) | <0.0001* |
| Ewekoro | 8.7 (2.0–38.0) | 0.004* | 9.7 (2.2–42.5) | 0.002* |
| Abeokuta South | 9.2 (2.2–39.5) | 0.003* | 7.8 (1.8–33.4) | 0.006* |
| Ifo | Ref | | Ref | |
| *Gender* | | | | |
| Male | 1.2(0.9–1.5) | 0.129 | 1.1(0.8–1.3) | 0.673 |
| Female | Ref | | | |
| *Age* | | | | |
| 5 -14y | 0.3 (0.2–0.4) | <0.0001* | 0.2(0.2–0.3) | <0.0001* |
| 15—24y | 0.8 (0.6–1.2) | 0.324 | 0.7 (0.5–1.0) | 0.084 |
| 25—34y | 1.3 (1.0–1.8) | 0.091 | 1.1 (0.7–1.5) | 0.761 |
| 35—44y | 1.1 (0.8–1.5) | 0.652 | 1.0 (0.7–1.4) | 0.829 |
| >44 | Ref | | Ref | |
| *Ivermectin treatment* | | | | |
| No | 0.2 (0.2–0.3) | <0.0001* | 1.5 (1.3–1.5) | <0.0001* |
| Yes | Ref | | Ref | |

participants from Odeda LGA Crude odd ratio (COR) = 112.8, Adjusted odd ratio (AOR) = 86.0). Infection risk is 52, 35 and 20 times higher among people from Abeokuta North (AOR = 52), Obafemi-Owode (AOR = 35.2), and Imeko-Afon (AOR = 19.6) LGAs respectively compared with Ifo LGA, while risk of infection was 8 and 10 times higher among people from Abeokuta South (COR = 9.2, AOR = 7.8) and Ewekoro (COR = 8.7, AOR = 9.7) (Table 3).

Those who are within the age range of 5-14years were 0.3 times likely to be infected with *O. volvulus*, while risk of infection was 0.8, 1.3, and 1.1 times higher in those who are within the age range of 15-24years, 25-34years and 35-44years respectively compared with those who are above 44years. Furthermore, risk of infection was 2-folds higher (COR = 0.2, AOR = 1.5) among those who did not receive preventive chemotherapy (ivermectin) compared with those who received (Table 3).

## Predicted risk of onchocerciasis in Ogun State

The predicted high-risk areas (probability > 0.8) of *O. volvulus* infection were all parts of Odeda, Abeokuta South, and Abeokuta North, southern part of Imeko-Afon, a large part of Yewa North, some parts of Ewekoro and Obafemi-Owode LGAs. The estimated prevalence for these regions were >60% (between 61% and 100%). The prediction showed that some part of Yewa North was expected to have high prevalence of onchocerciasis in contrast to the geospatial distribution with zero percent prevalence from the sampled location (which differs from the area predicted for high prevalence) (Fig 4).

The other parts of Imeko-Afon has between very low (0–10%) and moderate (21–40%) prevalence risk. Areas with very low to moderate predicted prevalence includes; Ipokia, Egbado South, Ifo, Ikenne, Shagamu, Remo-North, Ijebu North (with higher prevalence in areas sharing boundaries with adjoining highly endemic LGAs. Areas with very low (0–10%)

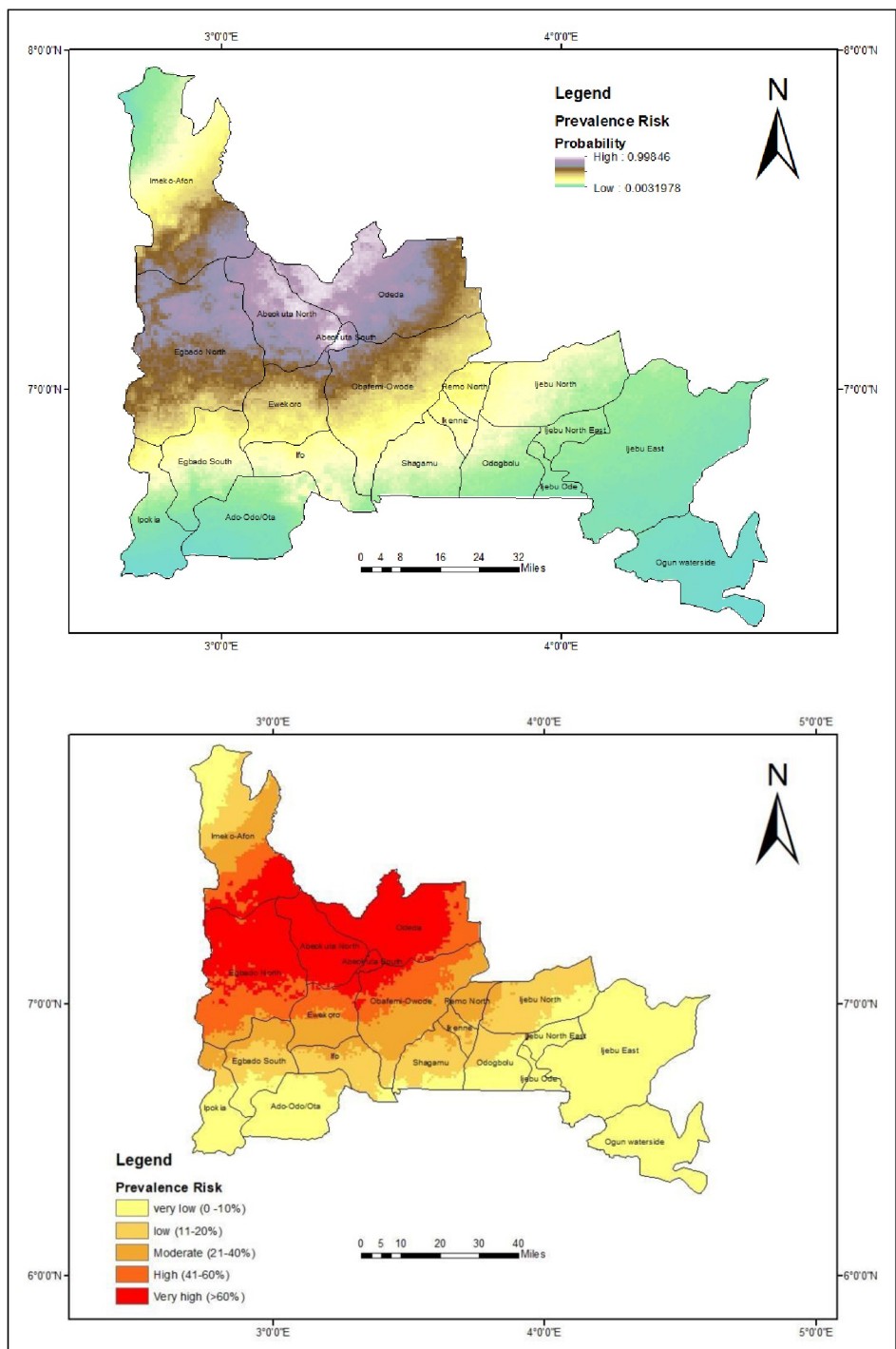

**Fig 4. Predicted prevalence of *O. volvulus* infection in Ogun State Nigeria.** This figure was created by the authors in R programming software (R version 4.1.2, Vienna, Austria). Available at https://www.R-project.org/. The Nigerian shapefile was obtained from World BankDataCatalog (https://data.humdata.org/dataset/geoboundaries-admin-boundaries-for-nigeria) an Open license standardized resource of boundaries (i.e., state, county) for every country in the world.

**Table 4. Average percent contribution and permutation importance of the variables used in the modeling of *O. volvulus* infection.**

| Environmental Variable | % Contribution | Permutation Importance |
|---|---|---|
| Bio15 (Precipitation Seasonality (Coefficient of Variation) | 53.6 | 49.7 |
| Bio12 (Annual Precipitation) | 43.5 | 43.0 |
| Bio19 (Precipitation of Coldest Quarter) | 1.7 | 7.3 |
| Bio6 (Min Temperature of Coldest Month) | 1.1 | 0.0 |
| Bio3 (Isothermality) | 0.2 | 0.0 |
| Bio8 (Mean Temperature of Wettest Quarter) | 0.0 | 0.0 |
| Bio2 (Mean Diurnal Range) | 0.0 | 0.0 |
| Bio1 (Annual mean temperature) | 0.0 | 0.0 |

to low (11–20%) prevalence risks include; Ado-odo ota (very low risk in major areas) and Odogbolu LGAs. Areas with very low prevalence risk are Ijebu North East, Ijebu East, Ijebu Ode, and Ogun waterside LGAs (Fig 4).

## Model performance and influencing factors

The average percent contribution (PC) and permutation importance (PI) of the 8 variables used in the modeling of onchocerciasis in this study were also assessed. In this study, precipitation seasonality (coefficient of variation) had the highest PC and PI of 53.6 and 49.7 respectively, followed by annual precipitation with PC of 43.5 and PI of 43.0 and precipitation of coldest quarter with PC of 1.7 and PI of 7.3. The results showed that both precipitation seasonality (coefficient of variation) and annual precipitation are strong predictors of *O. volvulus* distribution in Ogun State, accounting for 97.1% of the variations in distribution observed (Table 4).

The ROC curve obtained as an average of the 10 replications runs is shown in Fig 5, and specificity was calculated. The average and standard deviation of the AUC for the 10 replicate

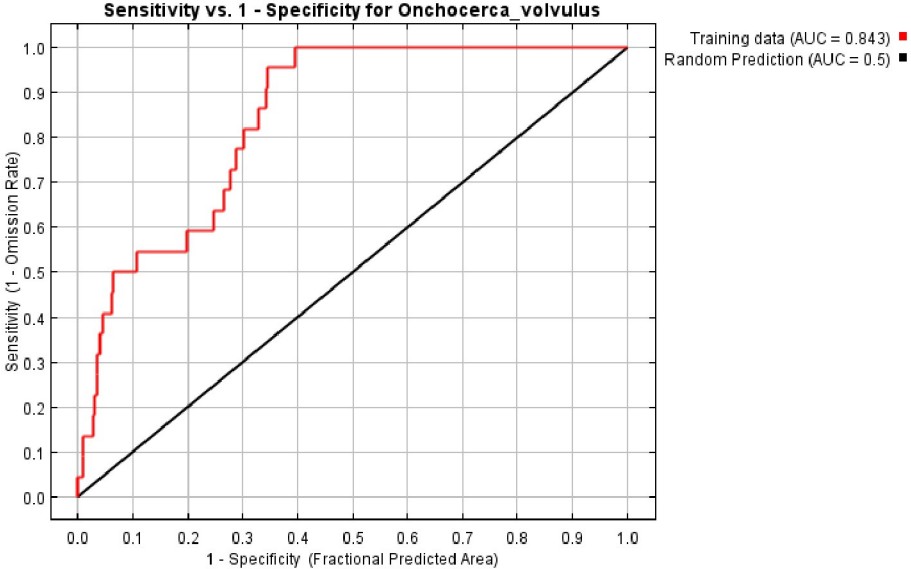

**Fig 5. Area under the curve (AUC) for *O. volvulus* prevalence.** Red line indicates the mean value for 10 MaxEnt replicate runs.

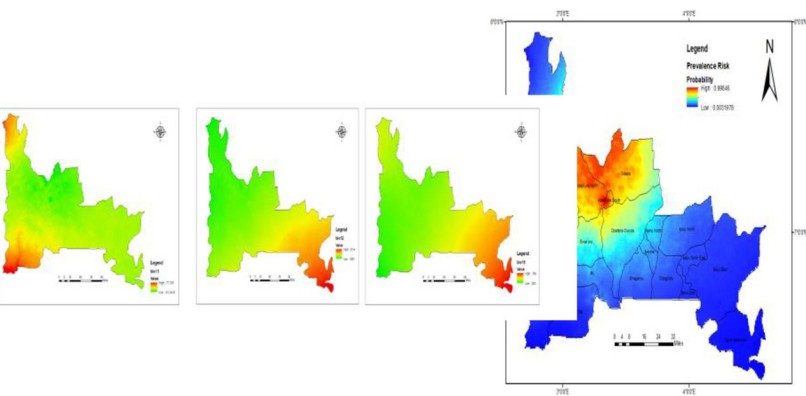

**Fig 6. The highest environmental variables that estimate to control the geographical distribution of *O. volvulus* in Ogun State.** Variable contributions (bio15, bio12 and bio19). This figure was created by the authors in R programming software (R version 4.1.2, Vienna, Austria). Available at https://www.R-project.org/. The Nigerian shapefile was obtained from World BankDataCatalog (https://data.humdata.org/dataset/geoboundaries-admin-boundaries-for-nigeria) an Open license standardized resource of boundaries (i.e., state, county) for every country in the world.

runs was 0.843±0.01. This value shows an excellent performance of the model as an AUC value of greater than 0.70 shows higher sensitivity and specificity for the presence of *O. volvulus*.

The relative importance of each variable to the prevalence of *O. volvulus* infection was also computed with the jackknife test in Fig 6 which gave a training gain of 0.65 and an area under the curve (AUC) value of 0.81 (red bar). The jackknife test also showed that bio15 (precipitation seasonality (coefficient of variation) and Bio 12 (annual precipitation) are the two variables that will affect the prevalence of onchocerciasis when used alone. The jackknife test also showed that bio15 (precipitation seasonality (coefficient of variation) decreased the gain the most when removed from the model.

Figs 6 and 7 shows the main highest estimated environmental variables (contributions) that determines the prevalence of *O. volvulus* infection in Ogun State. Spatial distribution analysis was done to determine the geographical variability with regards to the selected environmental variables in the State. The response curves of three variables to *O. volvulus* habitat suitability are shown in (Fig 8). We found out that precipitation seasonality (bio15) ranged from 61 to 77 mm, annual precipitation (bio12) ranged from 1901 to 2014 mm while precipitation of the coldest quarter (bio19) ranged from 323 to 704 mm. The response curves showed that between annual precipitation of 1100 and 1200 mm favors the potential transmission of *O. volvulus*. Similarly, precipitation seasonality of 60 to 64 mm significantly and potentially favoured the distribution of *O. volvulus* while precipitation of the coldest quarter ranging between 400 and 450 mm significantly favoured the transmission of *O. volvulus* in Ogun State (Fig 8).

The prediction model of prevalence risk was divided into 5 classes; very low (0–10%), low (11–20%), moderate (21–40%), high (41–60%) and very high (>60%). Ogun state has an estimated landmass of 16,980.55 km$^2$ in which potential distribution of *O. volvulus* infection cover an area of 10, 952.32 km$^2$ (about 64% of the total land mass). This area divided as; 2706 km$^2$ low prevalence probability, 2455 km$^2$ moderate prevalence probability, 2040 km$^2$ high prevalence probability, and 6028 km$^2$ very high prevalence probability.

## Discussion

In this study, we utilized *O. volvulus* infection data from a cross-sectional survey across eight-endemic LGAs in Ogun State Nigeria, to produce model-based estimates of infection risk for

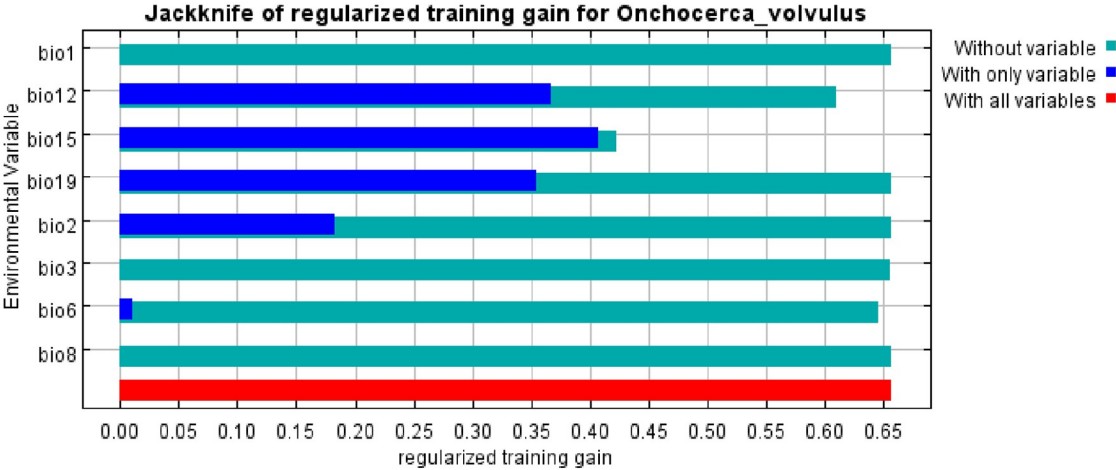

(a)

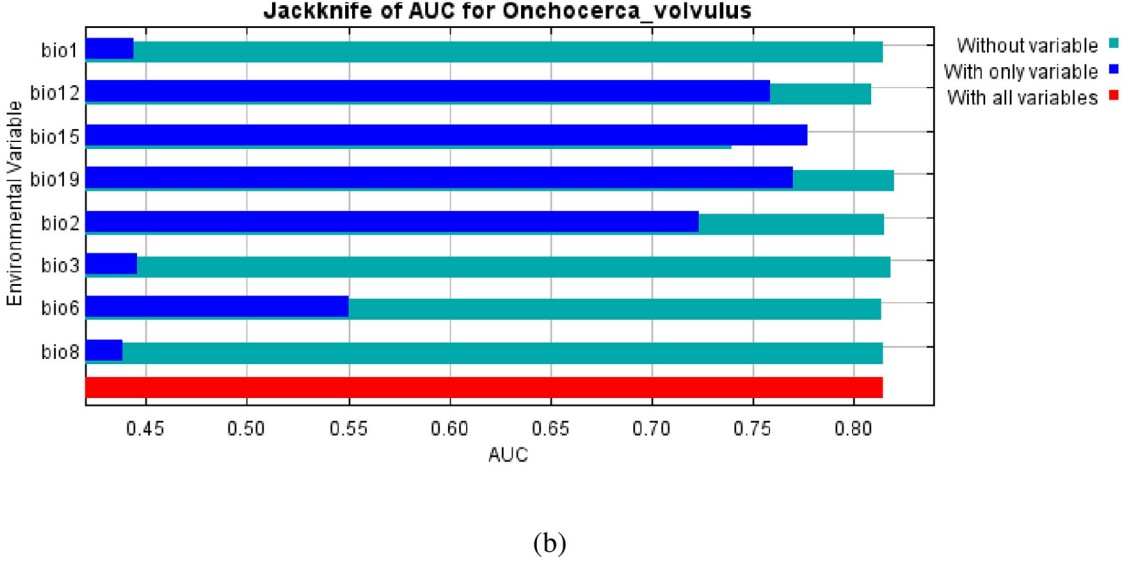

(b)

**Fig 7. Jacknife analysis (a) training gain (b) AUC.** The dark blue, light blue and red bars represent results of the model with each individual variable, all the remaining variables and all variables respectively.

the whole State. To the best of our knowledge, this is the first predictive geospatial-model for prevalence of *O. volvulus* infection in Ogun State, using empirical data.

The empirical prevalence of 9.3% recorded in this study, with higher prevalence in certain LGAs is an indication that elimination of onchocerciasis may still be a problem in Ogun State if control efforts are not geared up. The geospatial distribution of infection revealed that all communities in Odeda LGA (a peri-urban LGA) showed remarkably high prevalence compared with other LGAs having either low or zero prevalence at the community level. Our

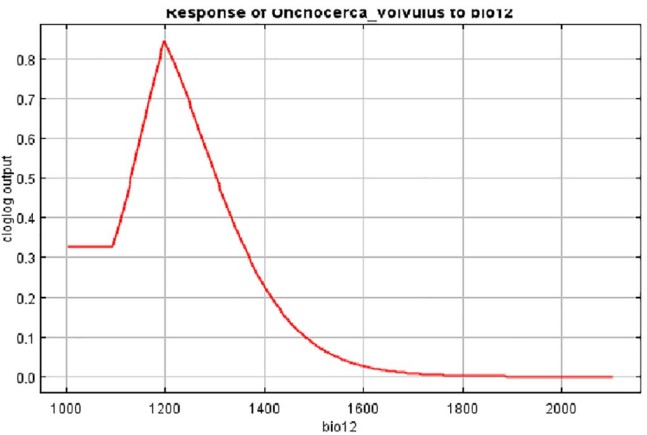

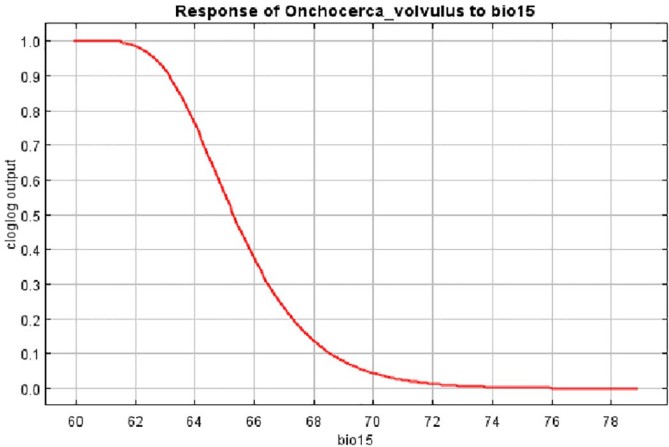

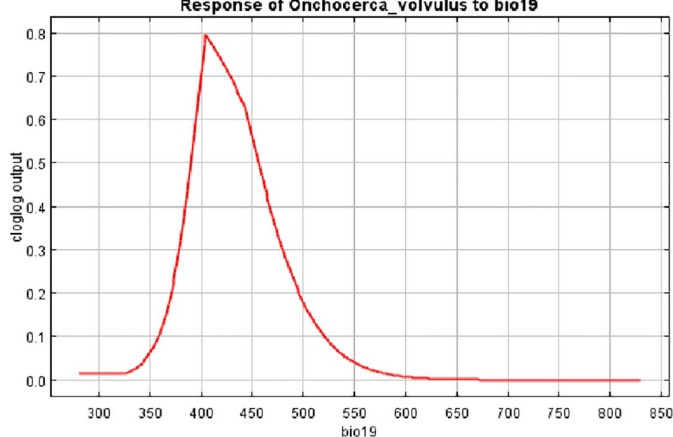

**Fig 8. Response curves of three environmental predictors used in MaxEnt model for *O. volvulus*.**

regression model further suggested an incredibly high risk of infection in Odeda LGA. This has serious implications for elimination. MDA has been on-going for over ten years in the state [7,38], it is quite worrisome that prevalence among children 5–14 years is still incredibly high or at par with adults across different LGAs in the State. This result showed that transmission is still actively on-going in the affected LGAs despite on-going MDA. This may be as a result of poor uptake of preventive chemotherapy with ivermectin observed among the study population. Our regression model also indicated a double-fold increase in risk of infection among those who did not take ivermectin compared with those who did. Studies have reported the efficacy of ivermectin in reducing mf load and subsequently reduce transmission [39,40]. Therefore, there is a need to investigate the reasons for this shortfall and how to subsequently improve geographical coverage if we are to meet the elimination target.

Modelling the baseline endemicity of infection is very germane in determining where and how to scale up control programmes. In order to raise adequate awareness and develop proper framework for advocacy in areas of community/institutional engagement, there is dire need to have good knowledge of the extent of disease burden [41]. In view of this, we used our model to predict the hotspots for onchocerciasis in Ogun State. It is important to note that predicted prevalence in locations that are not currently occupied by humans can increase risk of transmission when people with infection establish communities in such places [42].

Our model showed varying degrees of predicted prevalence across Ogun State. The predicted high-risk areas (probability >0.8) of onchocerciasis infections were all parts of Odeda, Abeokuta South, and Abeokuta North, southern part of Imeko-Afon, a large part of Yewa, some parts of Ewekoro and Obafemi-Owode LGAs. One thing that these LGAs had in common and could possibly explain the high prevalence recorded in this study is the availability of fast flowing rivers which is a potent breeding sites responsible for consistent bites by the *S. damnosum* s.l [7]. The higher risk of infections in Odeda communities, Abeokuta South, and Obafemi-Owode can be attributed to their proximity to Ogun river, while that of Abeokuta North and Yewa North is the proximity to Arakanga and Yewa river respectively. It is therefore expected to have higher risk of onchocerciais prevalence in communities along river system due to the preponderance of the *Simulium* vectors in these areas [43].

The prediction showed that some part of Yewa North was expected to have high prevalence of onchocerciasis in contrast to the geospatial distribution with 0 prevalence from the sampled location (which differs from the area predicted for high prevalence). This is quite possible and this calls for proper evaluation. Eight LGAs are known to be endemic for onchocerciais in Ogun State, however, a prevalence of 26.4% was reported by [44] in a community in Ijebu North East, a LGA that was thought to be non-endemic for onchocerciasis. It is therefore important to extend control programmes across all the LGAs if we want to effectively disrupt onchocerciais transmission in Ogun State. We also noticed in our predictive maps that some LGAs (that were previously considered as non-endemic) with high infection risk were from areas sharing boundaries with adjoining highly endemic LGAs. This suggest that when planning for control programmes, communities sharing boundaries with highly endemic LGAs should be given as much attention as possible if we are to drive the elimination of onchocerciasis in Ogun State.

It is equally important to stress, that our model revealed that transmission could have spread to additional eight LGAs in Ogun State, with low to moderate infection risk. This finding corroborates that of [45], who recently reported a prevalence of 1.64% in Yewa South LGA, which was presumably considered as non-endemic. The findings from this study should be taken as warning alert, requiring drastic and deliberate actions in other not to frustrate the existing and limited control efforts within the state.

Three variables; precipitation seasonality (coefficient of variation), annual precipitation and precipitation of coldest quarter were strongly associated with prevalence of onchocerciasis in

this study. A study by [46] also stressed the association between precipitation and prevalence of onchocerciasis. This finding suggests that optimum condition for breeding of blackflies and transmission of onchocerciais may be present in areas which the majority of precipitation occurs in the cooler parts of the year.

This study has predicted the prevalence of *O. volvulus* infection in Ogun State using a robust geospatial modeling approach, and as well the spatial pattern of disease spread. The empirical and predicted prevalence for *O. volvulus* infection was very high at some LGAs while infection may have spread to previously thought non-endemic LGAs. The empirical data used for this model was limited to 8 LGAs (32 communities) with 107 occurrence data points, hence may have present a limitation to this rigorousness of the model if the value for AUC is not up to 0.75. The MaxEnt modeling used in this study is a general-purpose method for making predictions of inferences from incomplete information [16,27]. The model used in this study is a present only modeling algorithm (i.e. absence data are not required). More so, the performance has been reported to be relatively better than other modeling methods [17,32]. The report that the model has been hardly influenced by small sample sizes with relatively robust prediction hence putting it among the top performing modeling tools [32] made it the choice model for our study.

## Conclusion

In Nigeria, the focus of onchocerciasis programs has shifted from control to elimination of *O. volvulus* transmission, prompting the need to employ newer tools to aid in efficient prioritization of decisions regarding elimination mapping and interventions [42] especially in areas that were not previously considered for intervention. As a result, we have generated a baseline prevalence map for Ogun State using geospatial modeling technique. Knowing that elimination mapping of any disease can be expensive, hence the map presented here could be used by the state onchocerciasis elimination programs to direct resources for elimination mapping. Coupled with that fact the method described here may be an inexpensive first step that can extrapolate state-wide prevalence from existing data, the national onchocerciasis elimination programs can also employ this mean in providing a better situational analysis of onchocerciasis in Nigeria from known prevalence data.

## Supporting information

**S1 File.**
(XLSX)

## Acknowledgments

We would want to express our gratitude to the community leaders and members who volunteered to take part in our research. We would also want to thank the health care professionals who, despite their hectic schedules, volunteered to assist us in this study.

## Author Contributions

**Conceptualization:** Olabanji Ahmed Surakat, Ayodele S. Babalola, Sammy O. Sam-Wobo.

**Data curation:** Olabanji Ahmed Surakat.

**Formal analysis:** Olabanji Ahmed Surakat, Ayodele S. Babalola, Monsuru A. Adeleke, Adedapo O. Adeogun.

**Investigation:** Sammy O. Sam-Wobo.

**Methodology:** Olabanji Ahmed Surakat, Monsuru A. Adeleke, Olufunmilayo A. Idowu, Sammy O. Sam-Wobo.

**Project administration:** Olabanji Ahmed Surakat.

**Resources:** Adedapo O. Adeogun.

**Supervision:** Adedapo O. Adeogun.

**Validation:** Monsuru A. Adeleke.

**Visualization:** Olabanji Ahmed Surakat, Ayodele S. Babalola, Olufunmilayo A. Idowu.

**Writing – original draft:** Olabanji Ahmed Surakat, Ayodele S. Babalola, Monsuru A. Adeleke, Adedapo O. Adeogun, Olufunmilayo A. Idowu, Sammy O. Sam-Wobo.

**Writing – review & editing:** Olabanji Ahmed Surakat, Ayodele S. Babalola, Adedapo O. Adeogun, Olufunmilayo A. Idowu, Sammy O. Sam-Wobo.

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
