## [Decision Letter · Decision Letter 0]

23 Aug 2022

PONE-D-22-16005Geospatial distribution and predictive modeling of Onchocerciasis in Ogun State, NigeriaPLOS ONE

Dear Dr. Olabanj Surakat,

Thank you for submitting your manuscript to PLOS ONE. After careful consideration, we feel that it has merit but does not fully meet PLOS ONE’s publication criteria as it currently stands. Therefore, we invite you to submit a revised version of the manuscript that addresses the points raised during the review process.

ACADEMIC EDITOR: The manuscript needs English language editing, beside the reviewers comments.

We look forward to receiving your revised manuscript.

Kind regards,

Shawky M Aboelhadid, PhD

Academic Editor

PLOS ONE

Journal Requirements:

- https://www.medrxiv.org/content/10.1101/2022.01.10.22269016v1.full

In your revision ensure you cite all your sources (including your own works), and quote or rephrase any duplicated text outside the methods section. Further consideration is dependent on these concerns being addressed.

6. We note that you have referenced (ie. Bewick et al. [5]) which has currently not yet been accepted for publication. Please remove this from your References and amend this to state in the body of your manuscript: (ie “Bewick et al. [Unpublished]”) as detailed online in our guide for authors

7. We note that Figures 1, 2, 3, 4 and 6 in your submission contain map images which may be copyrighted. All PLOS content is published under the Creative Commons Attribution License (CC BY 4.0), which means that the manuscript, images, and Supporting Information files will be freely available online, and any third party is permitted to access, download, copy, distribute, and use these materials in any way, even commercially, with proper attribution. For these reasons, we cannot publish previously copyrighted maps or satellite images created using proprietary data, such as Google software (Google Maps, Street View, and Earth). For more information, see our copyright guidelines: http://journals.plos.org/plosone/s/licenses-and-copyright.

a. You may seek permission from the original copyright holder of Figures 1, 2, 3, 4 and 6 to publish the content specifically under the CC BY 4.0 license.  

Reviewers' comments:

Reviewer's Responses to Questions

**Comments to the Author**

1. Is the manuscript technically sound, and do the data support the conclusions?

Reviewer #1: Yes

2. Has the statistical analysis been performed appropriately and rigorously? 

Reviewer #1: Yes

3. Have the authors made all data underlying the findings in their manuscript fully available?

Reviewer #1: No

4. Is the manuscript presented in an intelligible fashion and written in standard English?

Reviewer #1: Yes

5. Review Comments to the Author

Reviewer #1: I read the manuscript PONE-D-22-16005 entitled "Geospatial distribution and predictive modeling of onchocerciasis in Ogun State, Nigeria" with interest. In this study, the authors obtained georeferenced onchocerciasis infection data from a cross-sectional survey collected between March and July 2015 from locations in Ogun State, Nigeria. However, it could have been a very interesting work if communicated earlier, as the MDA interventions in the past seven years might have changed the epidemiological situation of the disease in the area. Though it is possibly losing its merit, this will provide useful information for identifying priority areas for delivery of intervention and eventually contribute to the elimination of onchocerciais in the study area.

The authors followed appropriate statistical methods and modelling to analyse the data. My comments on the manuscript are on the use of abbreviations and punctuation throughout the manuscript. There are many small errors in punctuation and improper use of abbreviations throughout the manuscript. For example, in the abstract portion, the abbreviation "LGA" should be written in its long form as it appears first, then in its abbreviated form. The same is true for the phrases "mass drug administration", "community-directed treatment with ivermectin", "Onchocerca volvulus" etc.. whereas WHO, GIS, RS, WHO-NTD, RDT, mf, AOR etc ... are written in abbreviated form in their first appearance in the manuscript. These and other terms in the manuscript have to be critically looked and corrected. If the manuscript is to be accepted, there may be a need for a language check by a professional editor for errors in punctuation in the manuscript.

Moreover, the authors need to follow the PLOSONE authors’ guide to format the manuscript, especially the ethical clearance and permission subsection, which needs to be placed before the result section of the manuscript.

6. PLOS authors have the option to publish the peer review history of their article (what does this mean?). If published, this will include your full peer review and any attached files.

Reviewer #1: No

---

## [Author Response · Author response to Decision Letter 0]

21 Nov 2022

. Please ensure that your manuscript meets PLOS ONE's style requirements, including those for file naming. The PLOS ONE style templates can be found at 

- https://www.medrxiv.org/content/10.1101/2022.01.10.22269016v1.full

Response: This has been adjusted and highlighted within the text (See lines 407-409 and 459-462)

In your revision ensure you cite all your sources (including your own works), and quote or rephrase any duplicated text outside the methods section. Further consideration is dependent on these concerns being addressed.

Response: This has been addressed, all sources have been cited within the text including my own works

Response: This was previously included in the manuscript (Line 199-208).

Response: The minimal data sets has been attached as a supporting information file in the revised submission

6. We note that you have referenced (ie. Bewick et al. [5]) which has currently not yet been accepted for publication. Please remove this from your References and amend this to state in the body of your manuscript: (ie “Bewick et al. [Unpublished]”) as detailed online in our guide for authors

Response: This has been amended to reflect unpublished 

7. We note that Figures 1, 2, 3, 4 and 6 in your submission contain map images which may be copyrighted. All PLOS content is published under the Creative Commons Attribution License (CC BY 4.0), which means that the manuscript, images, and Supporting Information files will be freely available online, and any third party is permitted to access, download, copy, distribute, and use these materials in any way, even commercially, with proper attribution. For these reasons, we cannot publish previously copyrighted maps or satellite images created using proprietary data, such as Google software (Google Maps, Street View, and Earth). For more information, see our copyright guidelines: http://journals.plos.org/plosone/s/licenses-and-copyright.

a. You may seek permission from the original copyright holder of Figures 1, 2, 3, 4 and 6 to publish the content specifically under the CC BY 4.0 license.  

Response: The maps were originally created by the authors; hence no permission was required. However, a statement indicating this has been added as a footnote to the maps. 

Reviewers' comments:

Reviewer's Responses to Questions

Comments to the Author

1. Is the manuscript technically sound, and do the data support the conclusions?

Reviewer #1: Yes

2. Has the statistical analysis been performed appropriately and rigorously? 

Reviewer #1: Yes

3. Have the authors made all data underlying the findings in their manuscript fully available?

Reviewer #1: No

4. Is the manuscript presented in an intelligible fashion and written in standard English?

Reviewer #1: Yes

5. Review Comments to the Author

Reviewer #1: I read the manuscript PONE-D-22-16005 entitled "Geospatial distribution and predictive modeling of onchocerciasis in Ogun State, Nigeria" with interest. In this study, the authors obtained georeferenced onchocerciasis infection data from a cross-sectional survey collected between March and July 2015 from locations in Ogun State, Nigeria. However, it could have been a very interesting work if communicated earlier, as the MDA interventions in the past seven years might have changed the epidemiological situation of the disease in the area. Though it is possibly losing its merit, this will provide useful information for identifying priority areas for delivery of intervention and eventually contribute to the elimination of onchocerciais in the study area.

The authors followed appropriate statistical methods and modelling to analyse the data. My comments on the manuscript are on the use of abbreviations and punctuation throughout the manuscript. There are many small errors in punctuation and improper use of abbreviations throughout the manuscript. For example, in the abstract portion, the abbreviation "LGA" should be written in its long form as it appears first, then in its abbreviated form. The same is true for the phrases "mass drug administration", "community-directed treatment with ivermectin", "Onchocerca volvulus" etc.. whereas WHO, GIS, RS, WHO-NTD, RDT, mf, AOR etc ... are written in abbreviated form in their first appearance in the manuscript. These and other terms in the manuscript have to be critically looked and corrected. If the manuscript is to be accepted, there may be a need for a language check by a professional editor for errors in punctuation in the manuscript.

Moreover, the authors need to follow the PLOSONE authors’ guide to format the manuscript, especially the ethical clearance and permission subsection, which needs to be placed before the result section of the manuscript.

Response: All punctuation errors have been addressed and abbreviations have been spelt out during first mention.

6. PLOS authors have the option to publish the peer review history of their article (what does this mean?). If published, this will include your full peer review and any attached files.

Do you want your identity to be public for this peer review? For information about this choice, including consent withdrawal, please see our Privacy Policy.

Reviewer #1: No

---

## [Decision Letter · Decision Letter 1]

3 Jan 2023

PONE-D-22-16005R1Geospatial distribution and predictive modeling of Onchocerciasis in Ogun State, NigeriaPLOS ONE

Dear Dr. Surakat, 

Thank you for submitting your manuscript to PLOS ONE. After careful consideration, we feel that it has merit but does not fully meet PLOS ONE’s publication criteria as it currently stands. Therefore, we invite you to submit a revised version of the manuscript that addresses the points raised during the review process.

ACADEMIC EDITOR: The abbreviations in the manuscript should be revised as recommended by the reviewers.

We look forward to receiving your revised manuscript.

Kind regards,

Shawky M Aboelhadid, PhD

Academic Editor

PLOS ONE

Journal Requirements:

Reviewers' comments:

Reviewer's Responses to Questions

**Comments to the Author**

1. If the authors have adequately addressed your comments raised in a previous round of review and you feel that this manuscript is now acceptable for publication, you may indicate that here to bypass the “Comments to the Author” section, enter your conflict of interest statement in the “Confidential to Editor” section, and submit your "Accept" recommendation.

Reviewer #1: (No Response)

2. Is the manuscript technically sound, and do the data support the conclusions?

Reviewer #1: Yes

3. Has the statistical analysis been performed appropriately and rigorously? 

Reviewer #1: Yes

4. Have the authors made all data underlying the findings in their manuscript fully available?

Reviewer #1: Yes

5. Is the manuscript presented in an intelligible fashion and written in standard English?

Reviewer #1: Yes

6. Review Comments to the Author

Reviewer #1: There are improvements from the previous version of the manuscript. However, the comments I raised in the previous review are not fully addressed. As mentioned earlier, most of the problem in the current version of the manuscript is with the use of abbreviations. There is improper use of abbreviations throughout the manuscript. The authors' have to critically look the manuscript for other similar errors and correct it accordingly.

7. PLOS authors have the option to publish the peer review history of their article (what does this mean?). If published, this will include your full peer review and any attached files.

Reviewer #1: No

While revising your submission, please upload your figure files to the Preflight Analysis and Conversion Engine (PACE) digital diagnostic tool, https://pacev2.apexcovantage.com/. PACE helps ensure that figures meet PLOS requirements. To use PACE, you must first register as a user. Registration is free. Then, login and navigate to the UPLOAD tab, where you will find detailed instructions on how to use the tool. If you encounter any issues or have any questions when using PACE, please email PLOS at figures@plos.org. Please note that Supporting Information files do not need this step.<quillbot-extension-portal></quillbot-extension-portal>

---

## [Author Response · Author response to Decision Letter 1]

13 Jan 2023

Response to reviewers comment

Line 31: The abbreviation "LGA" should be written in its long form as it appears first, then in its abbreviated form.

Response: The abbreviation has been re-written in the long form as Local Government Areas

Line 33: Please write onchocerca volvulus in its abbreviated form, i.e., "O. volvulus".

Response: This has been changed in all places it appeared as O. volvulus to O. volvulus

Line 55: Please write onchocerca volvulus in its abbreviated form, i.e., "O. volvulus" . Similar comments to lines 76, 142, 143, 144, 214, 252, 254, 261, 267, 268, and 397. 

Response: This has been changed in all places it appeared as O. volvulus to O. volvulus

 Lines 66 and 67: The phrase "community-directed treatment with ivermectin", should be written in abbreviated form, as it had already been abbreviated in line 61 of the manuscript.

Response: The phrase has been written in the abbreviated form

 Line 68 and 69: The phrase "mass drug administration" should be written as "MDA" as you have already started using the abbreviated form of it in line 58 of the manuscript. Also, similar comment to line 407 of the manuscript.

Response: The phrase has been written in the abbreviated form

Line 69: Write the abbreviated version of microfilaria in parantheses i.e. microfilaria. Also, use the abbreviated form of microfilaria in lines 97 and 127 of the manuscript.

Response: The abbreviated form of microfilaria has been written in parenthesis

Line 111: Use "LGAs" instead of "Local Government Areas".

Response: Local Government Areas has been re-written as LGAs

Line 118 and 119: Use the abbreviated form of Rapid Diagnostic Test i.e. "RDT".

Response: The abbreviated form of Rapid Diagnostic Test has been used

Line 150: Please use "19" instead of "nineteen".

Response: Line 150 has been corrected to reflect 19

Line 262: Abbreviate the "Crude Odd ratio" as "COR", then, start to use this abbreviation following it.

Response: Crude Odd ratio has been abbreviated to reflect as "COR”

Line 219: Please use "two" instead of "2".

Response: Line 219 has been corrected to reflect two

Line 221: Add "percent" next to "zero" i.e. zero percent prevalence.

Response: Percent has been added to zero

Lines 267 and 269: replace "y" with "years".

Response: Years has been used to replace y in all places it appears

Line 299: Please replace "0" with "zero percent".

Response: O has been replaced with Zero percent

Line 317: Please use the abbreviated version for "Receiver Operating Characteristics".

Response: The abbreviated version for "Receiver Operating Characteristics" has been replaced with ROC

Line 319: Please use the abbreviated version of "Areas under curve". Similar comment to lines 323 and 324.

Response: The abbreviated version for " Areas under curve " has been replaced with AUC.

Line 394: Replace "8" with "eight".

Response: 8 has been replaced with eight

Line 404: Please replace "it’s" with "it is".

Response: It’s has been replaced with it is

Line 426: Please use the abbreviated version of the species name. i.e. S. damnosum s.l.

Response: The abbreviated version of the species name has been used

Line 461 and 462: Please use the abbreviated form of "maximum entropy Modelling" only .i.e. "MaxEnt". 

Response: MaxEnt has been replaced with maximum entropy Modelling

---

## [Decision Letter · Decision Letter 2]

30 Jan 2023

Geospatial distribution and predictive modeling of Onchocerciasis in Ogun State, Nigeria

PONE-D-22-16005R2

Dear Dr. Surakat,

We’re pleased to inform you that your manuscript has been judged scientifically suitable for publication and will be formally accepted for publication once it meets all outstanding technical requirements.

Kind regards,

Shawky M Aboelhadid, PhD

Academic Editor

PLOS ONE

Additional Editor Comments (optional):

Reviewers' comments:

Reviewer's Responses to Questions

**Comments to the Author**

1. If the authors have adequately addressed your comments raised in a previous round of review and you feel that this manuscript is now acceptable for publication, you may indicate that here to bypass the “Comments to the Author” section, enter your conflict of interest statement in the “Confidential to Editor” section, and submit your "Accept" recommendation.

Reviewer #1: All comments have been addressed

2. Is the manuscript technically sound, and do the data support the conclusions?

Reviewer #1: Yes

3. Has the statistical analysis been performed appropriately and rigorously? 

Reviewer #1: Yes

4. Have the authors made all data underlying the findings in their manuscript fully available?

Reviewer #1: Yes

5. Is the manuscript presented in an intelligible fashion and written in standard English?

Reviewer #1: Yes

6. Review Comments to the Author

Reviewer #1: The authors have fully addressed my concerns and comments raised in the previous version of their manuscript.

7. PLOS authors have the option to publish the peer review history of their article (what does this mean?). If published, this will include your full peer review and any attached files.

Reviewer #1: No

<quillbot-extension-portal></quillbot-extension-portal>

---

## [Editor Report · Acceptance letter]

10 Feb 2023

PONE-D-22-16005R2 

Geospatial distribution and predictive modeling of Onchocerciasis in Ogun State, Nigeria 

Dear Dr. Surakat:

I'm pleased to inform you that your manuscript has been deemed suitable for publication in PLOS ONE. Congratulations! Your manuscript is now with our production department. 

Kind regards, 

on behalf of

Professor Shawky M Aboelhadid 

Academic Editor

PLOS ONE